# Psychosocial Features of Shift Work Disorder

**DOI:** 10.3390/brainsci11070928

**Published:** 2021-07-14

**Authors:** Annie Vallières, Chantal Mérette, Alric Pappathomas, Monica Roy, Célyne H. Bastien

**Affiliations:** 1École de Psychologie, Université Laval, Québec, QC G1V 0A6, Canada; alric.pappathomas.1@ulaval.ca (A.P.); makkita@yahoo.com (M.R.); Celyne.Bastien@psy.ulaval.ca (C.H.B.); 2Centre de Recherche CERVO, Québec, QC G1E 1T2, Canada; chantal.merette@psa.ulaval.ca; 3Centre de Recherche du Centre Hospitalier Universitaire de Québec-Université Laval, Québec, QC G1L 3L5, Canada; 4Département de Psychiatrie et de Neurosciences, Faculté de Médecine, Université Laval, Québec, QC G1V 0A6, Canada

**Keywords:** shift work disorder, insomnia, psychosocial variables, sleepiness, night work

## Abstract

To better understand Shift Work Disorder (SWD), this study investigates insomnia, sleepiness, and psychosocial features of night workers. The study compares night workers with or without SWD to day workers with or without insomnia. Seventy-nine night workers and 40 day workers underwent diagnostic interviews for sleep disorders and for psychopathologies. They completed questionnaires and a sleep diary for 14 days. The design was observatory upon two factors: Work schedule (night, day work) and sleep (good sleep, SWD/insomnia). Two-way ANCOVAs were conducted on psychosocial variables, and effect size were calculated. The clinical approach chosen led to distinct groups of workers. Night workers slept several periods (main sleep period after work, naps, nights on days off). High total wake time and low total sleep time characterized sleep in SWD. Most night workers with SWD still complained of sleepiness after main sleep. Cognitive activation distinguished groups of night workers. All other differences in psychosocial variables between night workers groups were similar to, but smaller than, the ones between day workers. The evaluation of SWD should consider all sleep periods of night workers with particular attention to self-reported total wake time, state sleepiness, and level of cognitive activation.

## 1. Introduction

Approximately one-quarter of the active population consists of shift workers [1] who work outside the traditional work times. They can be either night workers or evening workers, continuously working the same shift or intertwined with days off. They can also be rotating shift workers functioning under several combinations of work shifts. A large number of shift workers may be afflicted with shift work disorder (SWD) that is defined as the presence of insomnia and/or excessive sleepiness occurring in temporal relation to their work schedule [2,3], and accompanied by a reduced sleep time duration [2]. The overall prevalence of SWD is estimated at 26.5% [4]. The SWD estimated prevalence is two to five times higher than anxiety, insomnia, or depression prevalence in the general population [5,6,7]. In addition, SWD is associated with an impact on mental and physical health [8,9,10]. Considering the high prevalence of SWD and its significant consequences for mental and physical health, a more comprehensive understanding of SWD is imperative.

### 1.1. Diagnosing Shift Work Disorder (SWD): A Complex Task

Diagnosing SWD is a real challenge. Because of its many facets, this seemingly simple disorder is actually complex to assess. The complexity of SWD is first of all related to the work schedule particularities that comprise the work shift [11,12], speed of the rotation between shifts (one or two days rotation or weekly rotation) [13], and shift duration [14]. Consequently, shift workers are accustomed to sleep during several periods out of 24 h: The main sleep period and naps [15]. Each particularity of the work schedule has negative consequences for the sleep and health of shift workers. For example, long working hours are associated with poor health [16], and all types of work shifts are associated with poor mental health (especially unpredictable and irregular shift work) [17]. Thus, work schedules should be carefully considered to evaluate sleep context in SWD diagnosis.

The complexity of SWD also relies on its two opposite symptoms, insomnia and sleepiness [2,3], each being linked to a specific period of time. Insomnia occurs during the sleep period and has consequences during wake time. It is, thus, considered as a 24 h disorder [18]. On the contrary, sleepiness occurs during the wake time period and may have consequences for the sleep period, decreasing sleep latency and increasing slow wave sleep [19]. The two symptoms can, therefore, interfere with each other. Moreover, shift workers may present both symptoms, insomnia alone or only sleepiness [20], and receive the same diagnosis (SWD). Therefore, this heterogeneity may complicate the clinical evaluation and treatment choice offered.

The way insomnia and sleepiness are defined and assessed in SWD can be confusing. First, the ICSD-III [2] and DSM-5 [3] indicate that insomnia should occur in a temporal relationship with the work schedule; however, it is not specified during which sleep periods insomnia occurs in SWD. Moreover, insomnia criteria are not defined in the SWD section of ICSD-III [2] and DSM-5 [3]. Only a very few studies give a portrait of sleep variables in SWD [20,21,22]. Together these studies show that shift workers with SWD present a shorter self-reported total sleep time, longer sleep onset latency, and lower sleep efficiency than shift workers without SWD. Second, the sleepiness symptom in SWD might appear more obvious than the insomnia ones, since it is expected to be related to wake time during the night shift. However, shift workers are not only awake during their night shift, and can, therefore, base their complaint of sleepiness on another time period. Another problem with sleepiness is that, most of the time, it is assessed with the Epworth Sleepiness Scale (ESS [23]) (see [9,24]) and rarely focuses on several wake time periods, as demonstrated in some studies [20,21,22]. For instance, with the Stanford Sleepiness Scale [25], it has been shown that shift workers with SWD present high state sleepiness before and after a night shift [21]. Gumenuyk et al. [20,22] specify that these state sleepiness levels are higher for night workers with insomnia and sleepiness than those having only insomnia.

To circumvent the SWD complexity, it is essential to take into account the global sleep picture of shift workers along with the simultaneous assessment of insomnia and sleepiness during all periods of sleep and wakefulness.

### 1.2. Psychosocial Features of SWD

Currently, the scientific literature explains SWD by the circadian misalignment, due to the work schedule [26]. Shift workers attempt to sleep when their circadian rhythm favors wakefulness. However, SWD seems to be influenced by other variables, since some psychosocial features, like anxiety [10,27,28], and depression levels [9,27], have been identified as being related to SWD. Irregular bedtime schedules, going to bed stressed or angry, and anticipating worrying in bed have all been identified as potential risks for developing SWD [27]. Sleep reactivity, which is the vulnerability for experiencing sleep disturbances as a result of a life stressor [10,27,28], has also been linked to SWD. Cognitive variables, such as dysfunctional beliefs about sleep, related to worries and helplessness, as well as selective attention toward noise and worries, were shown to be part of SWD [29]. Other features, such as work satisfaction [12] and marital satisfaction [30], are related to shift work and could be considered as sleep facilitators during wakefulness. Although some psychosocial features have been identified to contribute to SWD, we still need to enlarge their investigation and to study their specificity in the shift work context in comparison to insomnia in the general population.

The investigation of psychosocial features of SWD should be based on an integrative model to capture the entire picture of SWD and consider its complexity. Although Cheng and Drake [31] suggested using an integrative approach to understanding SWD, no conceptual model has yet been applied to this disorder. The psychobiological model of sleep proposed by Espie [32] could be used for this purpose. The model proposes a combination of physiological and psychological components to explain sleep regulation. According to this model, sleep is a natural state produced by sleep regulation mechanisms (circadian rhythm and sleep homeostasis) as explained by the two-process model of sleep [33,34]. The psychobiological model adds that good sleep is maintained by four central and interactive components that combine psychosocial contributors: (a) Physiological de-activation; (b) cognitive de-activation; (c) sleep-stimulus control; and (d) sleep facilitation during wakefulness. This model also posits that emotions, which are regulated by cognitive and physiological activations, should be neutral to favor good sleep. The advantage of using this model for SWD to explain insomnia in the general population is that it fully integrates sleep regulation mechanisms, including circadian rhythm, which is an essential component in SWD.

### 1.3. Objectives of the Current Study

To deepen the understanding of SWD, the present study investigates the global sleep context of night workers with the psychobiological model of sleep [32]. The objective is to evaluate psychosocial variables that distinguish night workers with SWD and those without SWD, satisfied with their sleep. A clinical approach adapted to the context of night work is used to adequately diagnose workers suffering from SWD. It is expected that using a global sleep evaluation will lead to a clear distinction between night worker groups. We predict that night workers with SWD will present higher cognitive activation, anxiety, and depressive levels than night workers with good sleep. We expected that these differences will be larger for night workers than for day workers.

## 2. Methods

### 2.1. Study Design

A two-factor passive, observational design was used. Hospital night work schedules in the Quebec City area were structured by the employer into 14-day blocks. Each 14-day block includes 6–10 night shifts completing the 14 days with days off. For instance, when six night shifts are worked, the 14 days are completed with eight days off. There are two work schedule variations: (1) Consecutive night shift: Working six to 10 consecutive nights, followed by eight to four days off, respectively, over 14 days; (2) rotating night shift: Working 6–10 nights, alternating with eight to four days off over 14 days. The choice between the two is made by the employer and is based on work seniority. The first factor, “work schedule”, had two levels: Night and day work. The second factor, “sleep disorder”, had two levels: Good sleep or presence of SWD (for night workers) or insomnia (for day workers). Each participant was assigned to one of the four groups: (a) Night workers with SWD; (b) night workers with good sleep; (c) day workers with insomnia; or (d) day workers with good sleep.

A clinical approach was chosen to evaluate night workers. Sleep and state sleepiness were evaluated daily during the work schedule planned by the participants’ employer, as well as during day off periods. The Espie’s integrative model [32] is used to provide a conceptual framework guiding the selection of psychosocial variables to be investigated in relation to sleep in the shift work context. To achieve the objectives, night workers, with and without SWD, were compared to day workers, with and without insomnia. Day workers are used as a control group as sleep, and psychosocial features between individuals with insomnia and those with good sleep are well documented [35,36,37]. The inclusion of day workers provides three advantages: (1) To avoid concluding that differences between night workers are because of the work schedule, while it can be due to the sleep disorder; (2) to identify what is psychosocially specific to SWD; and (3) to discuss whether results found are similar, less, or larger than the ones known for day workers.”

### 2.2. Setting

Participants were recruited in hospitals in the Quebec City area (the CIUSSS de la Capitale-nationale and the CHU de Québec) by means of the internal email service, the internal newsletter of the CHU de Québec, and bulletin boards. Ethical approval was obtained from the CIUSSS de la Capitale-nationale ethics committee (#233) sector mental health and neurosciences and the CHU de Québec ethics committee (#CHS10-08-060). Recruitment was performed between 2010–2015.

### 2.3. Participants

Inclusion criteria for night workers were (a) being over 18 years old; (b) working at least six nights out of 14 days for at least three months; (c) night work had to be between 12 a.m. (midnight) and 8 a.m. (±1 h). Exclusion criteria were: (a) Working at least twice on an evening schedule from 4 p.m. to 12 a.m. (midnight) over 14 days; (b) working more than 9 h within the same shift; (c) presenting a possible sleep disorder other than SWD or insomnia (for day workers) (e.g., sleep apnea, periodic limb movement, bruxism, or parasomnia); (d) presenting a major depression with suicidal ideation; (e) presenting a psychotic disorder or any disorders resulting from substance abuse; (f) being unable to answer questions during interviews or to respond to questionnaires; and (g) being visually impaired. The inclusion and exclusion criteria were the same for day workers except that their work schedule had to be between 8 a.m. and 8 p.m. (±1 h), and they should not have worked during the night over the previous three months. No specific professional activity was targeted.

Prospective participants (348 individuals) completed a telephone screening for eligibility assessment. Among them, 150 were eligible and came to the sleep laboratory. Fourteen participants were excluded because they were either working evening shifts (*n* = 6), they reported having another sleep disorder (*n* = 4), they did not want to complete the sleep diary (*n* = 3), or they had more than nine hours in the same shift (*n* = 1). Another 17 people were excluded because it was impossible to diagnose SWD (or insomnia for day workers) or good sleep. The evaluation showed that these participants reported being dissatisfied with their sleep without excessive sleepiness or insomnia (according to the above inclusion criteria). Conversely, they were satisfied with their sleep, but had insomnia or sleepiness. Thus, these participants appear to have a sleep pattern that falls somewhere between insomnia and good sleep. In addition, being dissatisfied with one’s sleep indicates a possible identity of insomnia (for more information, see [38]) and would not be congruent with the definition of a good sleeper. Finally, they were excluded to prevent amplifying the variance and interfering with statistical analyses.

The full sample (N = 119) consists of night workers with good sleep (*n* = 36) or SWD (*n* = 43) and day workers with good sleep (*n* = 20) or insomnia (*n* = 20). The mean age of the sample was 37.8 years (SD = 12.1), and 77.3% were women. 66.4% of the sample were nurses or nursing assistants, 11.8% administrative agents, 6.7% various technicians, and 15% various other hospital workers, such as housekeeping employees, dietitians, research professionals, or psychologists. The number of years working under this schedule was higher for night workers with good sleep (7.9 years) than for night workers with SWD (4.1 years) (*p* = 0.0096). The groups were comparable for all other sociodemographic variables (see Table 1).

## 3. Measures

### 3.1. Initial Screening

Phone Screening. A 15-min phone screening to determine participant eligibility was conducted by two undergraduate psychology students. Sociodemographic data were collected, as well as home and work addresses, to determine the distance in kilometers from home to work.

Evaluation. The evaluation consisted of a combination of the Structured Insomnia Interview (SII [39] adapted to screen for SWD and of the Semi-structured Clinical Interview for DSM-IV (SCID-IV and SCID-5; [40,41]) to screen for psychopathologies. The evaluation lasted for a maximum of three hours and was conducted by three graduate psychology students who had completed the clinical evaluation training included in their program. They were under the supervision of a clinical psychologist with expertise in sleep (A.V.), who provided four hours of training to the interviewers before the study.

The SII [39]-adapted has been described previously [29], and consists of seven sections. The first three sections included the 13 questions regarding difficulties falling asleep, staying asleep, and waking up too early. Each section focused on a specific sleep period: Main sleep (day sleep after night work); naps, and night sleep (on days off; this section is equivalent to the original SII [39]. The perceived severity of the sleep difficulty, the sleep satisfaction, and the degree of distress associated with each sleep difficulty were also evaluated using three items from the Insomnia Severity Index [42], and are answered on a Likert scale from “0” (very satisfied) to “4” (very dissatisfied). Participants were considered to be satisfied when they answered “0” or “1”. A fourth section evaluated the level of sleepiness within seven items. The first item was “Do you suffer from excessive sleepiness; yes or no?”. The remaining six items assessed the level of state sleepiness before going to work, during work, after work, before going to bed, and after waking up. These five items were related to night work and day sleep for night workers, and to day work and night sleep for day workers. The sixth item is the level of state sleepiness during days off. Each item was answered according to the Stanford Sleepiness Scale (SSS; [25]) with a Likert scale from “1” to “7”, a higher score indicating a higher subjective feeling of being sleepy. A composite score of state sleepiness was obtained by computing the mean of the six items. A fifth section of the SII-adapted evaluated sleep routine before going to bed and the bedroom environment, while a sixth one evaluated other potential sleep disorders to control for the exclusion criterion (c). Finally, the last section comes from the Standard Shift Work Index [43], and assesses what type of work schedule the participant follows (the number of day, night, and evening shifts and the number of days off for the next 14 days).

### 3.2. Sleep and Sleepiness Measures

Sleep Diary. Participants completed paper sleep diaries after each sleep period every day. The sleep diary used [44] is based on the consensus sleep diary [45], but allows participants to indicate the duration of their sleep upon a 24 h line. After each sleep period, participants reported what time they went to bed, how long they took to fall asleep, how long they stayed awake, if they woke up, and what time they got out of bed. From these diaries, Total Sleep Time (TST), Total Wake Time (TWT), and Sleep Onset Latency (SOL) were derived for each participant for main sleep, night sleep, and napping. The TWT includes early morning awakening (EMA), wake after sleep onset (WASO), and SOL. As such, the TWT comprised the wake time between the time the participant went to bed with the intention of falling asleep and the time he or she got out of bed. For day workers, the same sleep variables were derived. Day workers’ main sleep may include night and day sleep, if any. Sleep variables were averaged out of 14 days regardless of work days or off days. The number of naps per day during days with naps and per 14 days were computed. State sleepiness measured by the SSS [25] was also recorded in the sleep diary before and after work, as well as before and after each sleep period.

Epworth Sleepiness Scale (ESS [23]). This scale includes eight items evaluating daytime trait sleepiness on a Likert scale (“0” to “3”). Total scores vary between “0” and “24”, a higher score indicates a higher probability of unexpectedly falling asleep. The psychometric properties of the French version are adequate [46].

### 3.3. Psychosocial Measures

Participants completed printed French versions of the questionnaires. Measures retained for the present study covered the components of the psychobiological model [32]: “physiological activation”, “cognitive activation”, “stimulus control-behavior”, “emotions”, and “sleep facilitation”.

#### 3.3.1. Physiological Activation

Predisposition Sleep Arousal Scale—somatic scale (*PSAS* [47]). The PSAS measures the intensity of somatic (PSAS-somatic; 8 items) and cognitive (PSAS-cognitive; 8 items) arousal. Participants rate each item on a “1” to “5” Likert scale. Scores indicate the intensity of arousal experienced, while going to sleep the previous night. The PSAS has adequate psychometric properties [47].

Consumption history. This in-house self-reported questionnaire includes one item to know if they are smokers (yes/no) and three items related to cigarettes, alcohol, and caffeine consumption per week. A glass of wine, beer, or liquor is considered a drink. The same rule applies to coffee and cigarettes, i.e., one consumption equal to one quantity regardless of the type of coffee or cigarettes.

#### 3.3.2. Cognitive Activation

Dysfunctional Beliefs and Attitudes about Sleep (DBAS-16 [48]). This scale includes 16 items measuring sleep-related cognitions on a “0” to “10” Likert scale. A high score indicates a strong endorsement of dysfunctional beliefs and attitudes concerning sleep. The DBAS-16 has adequate psychometric properties with day workers [48].

Predisposition Sleep Arousal Scale—cognitive scale (PSAS [47]). The cognitive scale is described above.

Glasgow Content of Thoughts Inventory (GCTI [49]). The GCTI comprises 25 items assessing the nature and frequency of intrusive thoughts at bedtime on a Likert scale from “1” to “4”. Higher scores indicate a high degree of intrusive thoughts at bedtime. The psychometric qualities of the French version have not yet been evaluated.

#### 3.3.3. Stimulus Control-Behavior

Sleep habits in bed (TV, phone, electronic tablet, and reading) are drawn from the section “bedroom environment” of the SII [39] adapted and answered by “yes” or “no”.

#### 3.3.4. Emotions

Beck Depression Inventory-II (BDI-II [50]). The BDI-II assesses symptoms of depression experienced during the preceding week. The 21 items are scored on a 4-point Likert scale (“0” to “3”). Higher scores suggest more severe depression symptoms. The psychometric properties of the French version are well-documented [50].

State-Trait Anxiety Inventory (STAI-State and Trait [51]). The STAI assesses anxiety as both a state and a trait. Each subscale has 20 items rated on a 4-point Likert scale, where “1” indicates “not at all” and “4” signifies “a lot”. Higher scores suggest more anxiety. The psychometric properties of the STAI are excellent [51].

#### 3.3.5. Sleep Facilitation

The next three questionnaires are from the Standard Shift-Work Index [43], which possesses good psychometrics qualities. The questionnaires were developed to be used independently and were translated to French.

Work satisfaction scale [43]. This questionnaire includes five items evaluating the level of work satisfaction, rated on a “1” to “7” Likert scale. A higher score indicates higher work satisfaction.

Work schedule satisfaction scale [43]. This questionnaire includes six items. The first two provide an overview of the work schedule, while the others assess the participant’s perceived level of control and satisfaction over his or her work schedule. A high score indicates high satisfaction.

Social and domestic satisfaction [43]. This questionnaire includes 18 items assessing satisfaction with the time the work schedule performs various domestic and social activities. A high score indicates high satisfaction.

Dyadic adjustment scale (DAS [52]). This questionnaire comprises 16 items evaluating the level of marital relationship distress. Each item is rated on a Likert scale (“0” to “5”). High scores indicate higher distress. The psychometric properties of the French version are well documented [53].

### 3.4. Procedures

After the 15-min telephone screening, eligible participants were invited to the laboratory, where they signed a written consent form and performed the evaluation. The SII [39] adapted performed during the evaluation was used to identify participants with a potential sleep disorder. To balance the sample, an *Apriori* match regarding gender and age was done. To facilitate recruitment, this procedure was abandoned after two years and replaced by a statistical control of gender and age. During the first visit, participants completed all questionnaires. Following the evaluation, participants maintained a sleep diary for two weeks. The research coordinator contacted each participant one week after the evaluation to ensure compliance with the procedures. After the two-week period, participants were invited again to the sleep laboratory. At that time, they received feedback and sleep hygiene advices. Financial compensation of $50 was given to participants at the end of the second visit, and a brief summary of their sleep data was sent to them in the following weeks. The assignation of participants to one of the four groups was done after the two-week period. Table 2 summarizes the study design, tools used, and procedures.

### 3.5. Participant Assignation

To conduct the assignation of each participant, a specific algorithm was developed based on the SWD criteria of the ICSD-III [2], the insomnia criteria of the DSM-5 [3]. The algorithm included insomnia DSM-5 criteria, which are (A) having one or more of the following symptoms: (1) Difficulty falling asleep (SOL > 30 min); (2) difficulty maintaining sleep (WASO > 30 min); (3) spontaneous early awakening (waking up earlier than expected without being able to go back to sleep); and (B) reporting distress with sleep difficulties. The first three sections of the SII-adapted [39] were used to determine whether participants met these criteria for any sleep period: Sleep after returning home (main sleep), the most difficult nap, and night sleep. For night workers, if insomnia was diagnosed using one of the sleep periods, then the insomnia criterion of SWD was met. To meet the sleepiness criterion, the composite score of state sleepiness had to be above 3. Participants were diagnosed with SWD if insomnia and/or the sleepiness criterion was met. Following the evaluation, the two-week sleep diary data were used to confirm if insomnia criteria were met. For day workers, insomnia criteria were assessed only for night sleep. Participants were referred to as having good sleep if they fell into the category “without SWD” or “without insomnia” (for day workers), reported being satisfied with their sleep (to ensure that good sleepers do not experience sleep distress), and presented an average main sleep time of at least five hours assessed with the sleep diary, the minimum advised in the treatment of insomnia [54].

### 3.6. Statistical Methods

A double data entry was performed by independent research assistants for the evaluation, sleep diaries, and self-reported questionnaires to maximize the integrity of the results. Data banks were compared and errors corrected, where applicable. Thereafter, the normality of data, as well as missing or extreme data, were checked by a certified statistician under standardized procedures [55]. Statistical analyses were conducted with SAS/STAT software, version 9.4 for Windows [56].

Descriptive statistics of sociodemographic variables were computed to describe the sample and the four groups of workers. Within each work schedule (night and day work), participants with a sleep disorder were compared to those with good sleep on sociodemographic variables using t-tests for continuous variables (age, years working under this schedule, schooling in years) and chi-square tests for categorical ones (gender, nationality, civil status, full time employment, annual income, residence). Descriptive statistics were performed regarding the number of night workers with insomnia per sleep period among workers with SWD who present insomnia symptoms. Descriptive statistics were also conducted regarding the presence of state sleepiness (evaluation) across different times for night workers with SWD who present sleepiness symptoms.

Two-way ANCOVAs were conducted with the factor sleep disorder (good sleep vs. SWD or insomnia) and the factor work schedule (night vs. day work), plus the interaction between sleep disorder and work schedule adjusted for covariates age and gender of participants. To confirm group distinction, the ANCOVA was performed for each of the sleep variables. To answer the main objective, the ANCOVA was performed for each psychosocial variable. The GLM procedure of the software SAS/STAT was used for the continuous variables. For the categorical variables, such as smoker or reading in bed, the GENMOD procedure was used with options DIST = poisson and LINK = log to obtain relative risks (RR). When the p-value for the interaction term was significant, Effect Size (ES) or RR comparing Good Sleep and SWD or Insomnia were calculated for night shift and day workers separately. When the interaction term was not significant, ES or RR was calculated for the whole sample of workers. Conventional benchmarks for interpreting small, medium, and large effect sizes are 0.10, 0.30, and 0.50 [57].

## 4. Results

### 4.1. Description of Groups Following the Evaluation

#### Symptoms of Shift Work Disorder (SWD)

For each of the night workers with SWD, the evaluation identified whether symptoms of insomnia and/or symptoms of sleepiness (evaluation) were present. Among the 43 participants with SWD, 55.8% presented only symptoms of insomnia, 11.6% presented only sleepiness symptoms (evaluation), and 32.6% showed both symptoms during the evaluation.

The Venn diagram presented in Figure 1 shows the distribution of night workers with SWD that includes insomnia symptoms (*n* = 38) across each sleep period. It indicates that for 32 participants, insomnia is present during the main sleep period. The overlapping circles indicate that 25 of these 38 participants also presented insomnia either during night sleep (14) or during a nap (4), or both (7).

The Venn diagram presented in Figure 2 shows the distribution of night workers with SWD that includes state sleepiness symptoms (evaluation) (*n* = 19) across different times. It indicates that, for 17 participants, sleepiness during the evaluation is present at bedtime of the main sleep period, while 10 are still sleepy upon waking. The overlapping circles indicate that 15 of these 19 participants also presented state sleepiness at least two times during the day, including after night work and at bedtime.

### 4.2. Sleep and Wake Time Variables

Table 3 shows the least squared means and standard deviations of the total sleep time (TST), total wake time (TWT), and sleep onset latency (SOL) for each of the main and night sleep periods according to work schedule and sleep disorder. For the main sleep period, the ANCOVA global model was significant for each sleep variable (main-TST, main-TWT, main-SOL), Fs(5, 113) = 9.13, 16.92, and 11.20, *p*s < 0.0001, respectively. For main-TST, no interaction between sleep disorder and work schedule was detected. Thus, the difference between workers with good sleep and those with SWD or insomnia was the same for day and night workers with an overall effect size (ES) of −0.56 (large effect). Significant interactions were detected for main-TWT and main-SOL, Fs(1, 5) = 18.48 and 24.5, *p*s < 0.0001, respectively. Night workers with SWD had higher TWT than those with good sleep (ES = 0.70, large effect). This difference is larger for day workers (ES = 2.38, large effect). Only day workers with insomnia take significantly more time to fall asleep than day workers with good sleep (ES = 2.10, large effect).

For the night sleep period, the ANCOVA global model was significant for each sleep variable (night-TST, night-TWT, and night-SOL), Fs(5, 113) = 3.2, 11.0, and 5.4, *p*s = 0.01, < 0.0001, and < 0.0001, respectively. For night-TST, no interaction was detected. Thus, the difference between groups was the same for day and night workers (ES = −0.42, medium effect). For night-TWT and night-SOL, a significant interaction was detected, Fs(1, 5) = 7.70 and 13.70, *p*s = 0.006 and <0.0001, respectively. Night workers with SWD had higher night-TWT than those with good sleep (ES = 0.81, large effect), but this difference is larger for day workers (ES = 1.90, large effect). For night-SOL, day workers with insomnia took significantly more time to fall asleep than day workers with good sleep (ES = 1.53, large effect).

### 4.3. Naps and Sleepiness

Table 4 presents the least squared means and standard deviations for the number of naps (number of additional sleep periods per day of naps and number of additional sleep periods during the 14 evaluation days), nap sleep and wake time, and sleepiness variables (ESS questionnaire and SSS and excessive sleepiness [% yes] from the evaluation) according to work schedule and sleep disorder. The ANCOVA global model was significant for the average number of additional sleep periods per day of naps, Fs(4, 93) = 5.81, *p* < 0.0001, for nap-TST and nap-TWT, Fs(5, 96) = 4.74 and 3.56, *p*s = 0.0007 and 0.0055, as well as for 14 days nap-TST and nap-TWT, Fs(5, 96) = 18.12 and 6.93, *p*s < 0.0001. The ANCOVA global model was also significant for sleepiness assessed with the SSS, Fs(4, 114) = 3.96, *p* = 0.005. Other measures of sleepiness and interactions were not significant. Night workers napped more than day workers, but the difference between workers with good sleep and those with SWD or insomnia was the same (ESs = −1.08 and −0.51, large effects). Over the 14 day period, night workers sleep more and have more wake time than day workers during their naps (ESs = −1.55 and −1.02, large effects). For state sleepiness, workers with SWD or insomnia are sleepier than good sleepers (ES = 0.60, large effect).

Appendix A presented the timing nap distribution over a 24 h period for night workers. Two periods of naps per day are highlighted: One occurring between 8 p.m. and 11 p.m. before going to work and one at work (between 3 a.m. and 5 a.m.). For the 19 participants identified as experiencing excessive sleepiness during the evaluation, none of them reported in their sleep diary high state sleepiness during work, and the only one reported high state sleepiness on days off. According to the sleep diary, 35.9% of their awakenings following the main sleep period were accompanied by high state sleepiness. This percentage rose to 60.4% when awakening from a nap. Similarly, based on the sleep diary, 36.1% of night work started with high state sleepiness, and 49.4% ended with high state sleepiness.

### 4.4. Differences in Psychosocial Variables

Table 5 presents the least squared means and standard deviation of the physiological and cognitive activations, stimulus control behavior, emotion, and sleep facilitation variables according to work schedule and sleep.

#### 4.4.1. Physiological Activation

The ANCOVA global model was significant for the physiological activation before going to bed (PSAS-somatic), F(5, 112) = 3.50, *p* = 0.006. A significant interaction was also detected, F(1, 5) = 4.80, *p* = 0.031, indicating that only day workers with insomnia had higher physiological activation before going to bed than day workers with good sleep (ES = 1.11, large effect). There was no significant difference for cigarettes, caffeine, and alcohol consumption per week amongst groups.

#### 4.4.2. Cognitive Activation

The ANCOVA global model was significant for the cognitive activation level reported before going to bed (PSAS-cognitive), dysfunctional beliefs and attitudes about sleep (DBAS-16), and intrusive thoughts at bedtime (GCTI) across the group, F(5, 111) = 6.10, 12.82, and 5.96, *p*s < 0.0001, respectively. No interaction was detected for the PSAS-cognitive and GCTI, suggesting that the difference between workers with good sleep and those with SWD or insomnia were the same for day and night workers (ESs = 0.90 and 0.99, large effects). The interaction was significant for dysfunctional beliefs and attitudes about sleep, F(1, 5) = 10.68, *p* = 0.001, indicating that the endorsement of dysfunctional beliefs and attitudes about sleep was higher for night workers with SWD than their counterparts with good sleep (ES = 0.70, large effect), this difference is larger in day workers (ES = 1.99, large effect). Appendix A illustrates the interaction effect for these variables.

#### 4.4.3. Sleep Stimulus Control Behavior

There is no significant difference among groups for these variables.

#### 4.4.4. Emotions Variables

The ANCOVA global model was significant for depression, anxiety state, and anxiety trait, Fs(5, 110) = 7.85, 3.63, and 7.07, *p*s < 0.0001, 0.004, and <0.0001, respectively. No significant interaction among groups was detected. The difference between these variables among individuals with good sleep and those with an SWD or insomnia were the same for day and night workers (ESs = 0.92, 0.66, and 0.90, large effects).

#### 4.4.5. Sleep Facilitation Variables

The ANCOVA global model was significant for only two variables in this section: Work satisfaction, F(4, 114) = 2.51, *p* = 0.046, and work schedule satisfaction, F(4, 98) = 8.81, *p* < 0.0001. No significant interactions were detected among groups suggesting that the difference between work satisfaction and work schedule satisfaction among good sleepers and participants with SWD or insomnia was the same for day and night workers (ESs = −0.53 and 1.16; large effects).

## 5. Discussion

The study provides a clear view of sleep and psychosocial variables related to SWD. The global sleep evaluation we undertook led to the proper distinction between night workers with SWD and those with good sleep. Importantly, our findings highlight that cognitive activation is a key variable to understanding SWD. The differences in all other psychosocial variables between night workers with SWD and those with good sleep were similar, while still smaller than the ones between the two groups of day workers. Furthermore, the study shows how night workers adapt their sleep to their work schedule as they slept during several periods out of 24 h. These periods are the main sleep period after work, supplemented by a nap before going to work and/or at work. On days off, night workers sleep at night. Insomnia might have occurred during any of these sleep periods, and most participants experience insomnia during more than one sleep period. Night workers and day workers differ on time to fall asleep, number of naps, and sleep and wake times during nap. Sleep onset latency is higher only for day workers with insomnia, while both groups of night workers nap more than day workers. Most of the night workers with SWD presenting excessive sleepiness are still sleepy after their main sleep.

### 5.1. Diagnosing SWD

The findings highlight that the evaluation of all the sleep periods of night workers, using the DSM-5 [3] insomnia criteria, identify insomnia related to the work schedule. In doing so, our study raises several seminal issues. First, both ICSD-III [2] and DSM-5 [3] should be more precise about when insomnia symptoms may occur in SWD, since in their current definitions of SWD, not all these sleep periods are covered. Second, assessment tools need to address each sleep period, whether they be interviews, questionnaires, or sleep diaries. Some studies have modified questionnaires to address day and night sleep [20,21,58]. Third, evaluating naps was important as night workers take one or two naps per day for an average of around 90 min, which represents around 20% of their total sleep time. Fourth, insomnia in SWD appears appropriately captured by total wake time for each sleep period. Each wake time variable (SOL, WASO, EMA) taken separately does not seem high enough to distinguish between night worker groups (see WASO in [20]). Indeed, the sleep latency is low for night workers and only distinguishes day worker groups. Intense sleep deprivation may favor falling asleep quickly, due to the sleep homeostasis process regardless of the circadian phase [59]. In the context of night work, it is likely that the work schedule may create sleep deprivation, thus explaining sleep latency results. The two other wake time variables (WASO, EMA) could be low if taken separately, as night workers with SWD are less likely to spend time in bed, while awake during the day than day workers do while awake at night. The night work context can offer various daily activities for workers who have difficulties sleeping during the day, with a circadian phase favoring wakefulness. This is unlike day workers who have insomnia during the night and who are restricted from a variety of nocturnal activities, while their circadian phase favors sleep. Larger effect sizes for total wake time in day workers, as compared to night workers might reflect this explanation. Finally, the evaluation underlines that in SWD, the total sleep time is not only problematic for the main sleep period, but also for the night sleep period.

Night sleep is intrinsically linked to the sleep context of shift workers. All night workers sleep at night on days off in our study, which concurs with the night workers’ preferences reported in Petrov et al. [60]. About half of those with SWD have sleep difficulties at night on days off and during the main sleep period after work (see Figure 1). These results are consistent with two other studies comparing night workers with SWD and without SWD [20,21], and one comparing day and night workers with insomnia [58]. This might reflect a partial adaptation of biological rhythms observed in night workers [61,62] and a lower level of melatonin [61]. The night workers that have insomnia only at night on days off (*n* = 3; Figure 1), might be chronobiologically entrained to sleep in the morning rather than at night, as it appears to be possible [22,63]. Nevertheless, in these two cases (partial or complete re-entrainment), insomnia would be related to the work schedules. Returning to night sleep on days off may lower circadian adaptation for night workers to sleep during the day. However, since most night workers choose this sleep strategy, night sleep (or any sleep period on days off) should be integrated into the evaluation procedure of SWD.

State sleepiness levels seem to be important to identify night workers with SWD who complain of excessive sleepiness. State sleepiness is built on a basal sleep drive (trait sleepiness) and can be influenced by environmental (e.g., work schedule) and personal factors (e.g., previous sleep duration) [19]. We used three assessment tools to evaluate sleepiness, a dichotomous question assessing excessive sleepiness in the SII [39] adapted, the SSS [25], assessing state sleepiness, and the ESS [23], which target daytime trait sleepiness. The dichotomous question and the ESS do not distinguish between night workers with SWD from those with good sleep. The SSS from the sleep diary shows that those with a complaint of excessive sleepiness present high state sleepiness at least twice out of 24 h, and most of them are still sleepy after main sleep. Therefore, it seems that a graded scale targeting short-term changes in state sleepiness during the night worker’s main wakefulness period is a useful tool for identifying night workers with SWD complaining of sleepiness. This result converges with two studies [21,22] that used the Karolinska Sleepiness Scale [64], another graded scale evaluating the state sleepiness.

### 5.2. Psychosocial Features of SWD

Results on psychosocial variables pointed out that cognitive activation before going to bed is a key variable to understanding SWD. Moreover, work and work schedule satisfaction, as well as anxiety and depression levels, might contribute to SWD. According to Espie’s model [32], we could predict that the work schedule imposed by night work contributes to a circadian rhythm misalignment which in turn would disrupt sleep automaticity and plasticity. SWD would then be maintained because the night worker appears unable to decrease their cognitive activation to facilitate sleep, while having high physiological activation. Physiological activation levels were high for both night worker groups, while they were high for day workers with insomnia only, and thus, does not seem a key indicator of SWD. Cigarette, caffeine, and alcohol consumptions were not significant, which is congruent with some other studies on SWD [21,28,65,66]. These consumptions may be different in other samples and should be assessed in other studies in relation to physiological activation. Nevertheless, as groups in our study were similar, these consumptions cannot contribute to the elevated physiological activation assessed. Therefore, the physiological activation of night workers may represent the effect of a circadian misalignment, as in Papantoniou et al. [61]. It seems that night workers with good sleep can engage in sleep because they achieve cognitive de-arousal. The high level of cognitive activation in night workers with SWD converges with results obtained by Gumenyuk et al. [20] using evoked potential response. The high level of cognitive activation indicates that night workers with SWD are mentally alert, as are the day workers with insomnia. Cognitive activation can also be present, while the circadian rhythm is partially or completely adjusted, as this adjustment is known to be possible [26].

Cognitive activation implies that the person is thinking, but it is not informative about the content of their thoughts. According to Harvey [67], cognitive tone activity and worries are linked to insomnia. For day workers, it has been demonstrated that negative work rumination is linked to insomnia [68], while for night workers, beliefs related to worries and helplessness, as well as selective attention toward noise and worries are linked to SWD [29]. Anxiety and depression levels were higher for workers with SWD or insomnia. These results are consistent with previous studies on SWD [10,28] and with studies on insomnia in the general population (see review [69]). High anxiety and depression levels might feed the cognitive activation level as explained in Espie’s model [32] by influencing the negative content of thoughts. Therefore, thoughts and ruminations (either worries or depressive negative ruminations) about work might be mediators between work and work schedule satisfaction and insomnia in SWD, which give greater importance to night work, related components in relation to sleep. It may be possible that under different work schedules (e.g., evening, rotating) or work types (e.g., police officer, truck drivers), the content of negative thoughts or the type of worries differ.

Stimulus control behavior variables were the same for the four groups of workers, and thus, do not seem to contribute to SWD nor to insomnia in our sample. Staying in bed, while awake seems, therefore, to have less impact on insomnia than expected. It might reflect the context of night work that offers various daily activities for those presenting sleeping difficulties during the day. However, it also questions the stimulus control mechanism in insomnia, as others have done [70] soon after the original stimulus control publication [71].

Within the sleep facilitators’ variables, work satisfaction is lower for workers with SWD or insomnia regardless of their work schedule. It appears that work satisfaction is important to sleep well and could be evaluated in other studies on SWD. Marital distress was not significant, indicating that it was not related to night work or to SWD. One other study found that shift work reduces marital quality [30]. However, as they did not make a comparison with day workers, it was impossible to clarify if these variables were specific to the work schedule or to the sleep disorder.

The proposed integrative model [32] applied to SWD can now be investigated further using the variables identified as significant. Doing so would have several benefits for the understanding of SWD. It would connect the current knowledge on the circadian misalignment in SWD to psychosocial factors that contribute to SWD. Ultimately, it would be useful for identifying new targets, expanding treatment options, and supporting studies testing cognitive behavioral therapy for SWD (e.g., [72]).

### 5.3. Strengths, Limitations, and Direction for Future Research

One of the purposes of this study was to evaluate night workers considering their daily sleep reality related to their work schedule. We chose a clinical approach to achieve this objective, which consisted of using assessment tools that would be available for most therapists. The methodology used provides clearly distinct groups of workers, reinforcing the identified psychosocial characteristics. We also chose to recruit in hospitals to increase the homogeneity of work schedules and types of work within the sample. Doing so created two limitations: One is related to the fact that our sample is mainly composed of women, representing the reality in Quebec hospitals [73]; The other pertained to the fact that consecutive night workers and rotating night workers were considered together in the present study. Results might be different if we compared consecutive vs. rotating night workers. Thus, these night workers should be compared in future studies as they may not display the same psychosocial characteristics. It would also be useful to understand insomnia and sleepiness in polyphasic sleep of night workers by assessing circadian rhythm alignment and sleep architecture. Moreover, it is likely that those having insomnia in several sleep periods have additional negative health consequences. Future studies should compare night workers having insomnia is only one sleep period to those having insomnia in two or three sleep periods. Another limitation pertains to the fact that we excluded participants not clearly good sleepers or affected with SWD. However, we believe that increasing the contrast between groups improves the psychosocial picture we obtained. Vanttola et al. [21] made the same choice. One limitation of our study is based on the self-reported measures we used. These tools are subject to measure reactivity which may create spontaneous changes or completion omissions. Using other devices, such as electronic sleep diaries or actigraphy, to assess sleep daily may also enhance future studies. Another improvement that could be done involves increasing the number of adapted and validated sleep assessment tools for the population under study, while taking into account main sleep, night-time sleep, and naps for shift workers. Several concepts related to the workplace, such as negative work rumination or work incivility, have been linked to sleep in workers [68,74], and they should be specifically studied within the population of night workers. Furthermore, in the next step, mediation analyses could be used in other studies to confirm the integrative model for SWD. Finally, the methodology used in the present study compelled the recruitment of individuals in each of the four groups. Therefore, the proportion of night workers with SWD and day workers with insomnia is not a prevalent rate of their respective sleep disorders in the target population.

## 6. Conclusions

In summary, the present study adds to the understanding of SWD by demonstrating psychosocial variables that distinguish night workers with good sleep from those with SWD. Moreover, our study is innovative by choosing a clinical approach to understanding SWD, bringing together the fields of insomnia and shift work research, and proposing a conceptual model to study these variables in SWD. The SWD definition could be enhanced by using a global complaints’ score of wake time in at least one of the reported sleep periods, accompanied by a shorter main sleep time and/or night sleep time averaged from the entire work schedule. State sleepiness, especially after main sleep, might also be a good indicator of SWD for those who complain of excessive sleepiness. Cognitive arousal before bedtime in terms of intrusive thoughts or endorsement of beliefs about sleep, seems to be key psychosocial variables to understanding SWD. Night workers psychologically adapted to night work are probably those who can achieve a cognitive de-arousal in the presence of a physiological activation. As work satisfaction is related to SWD and insomnia, it is imperative to evaluate the content of thoughts, as well as ruminative thinking, to evaluate their relationship with work satisfaction and high cognitive activation. It would also be interesting to ascertain whether night workers with good sleep are chronobiologically re-entrained, at least partially, and if their sleep homeostat is still functioning adequately. This information, combined with knowledge about sleep and contributing psychosocial variables, would allow for interventions to be developed and adapted to night workers. In that respect, studies on cognitive behavior therapy for insomnia (CBT-I) for shift workers are already published [72,75,76], while our group is currently adapting CBT-I to the SWD targeting all sleep periods of shift workers based on our pilot studies [77,78]. Finally, the present study brings hope to night workers: It is possible to work at night and be satisfied with sleep. This could then become the goal of night workers with an SWD seeking help for their sleep.

## Figures and Tables

**Figure 1 brainsci-11-00928-f001:**
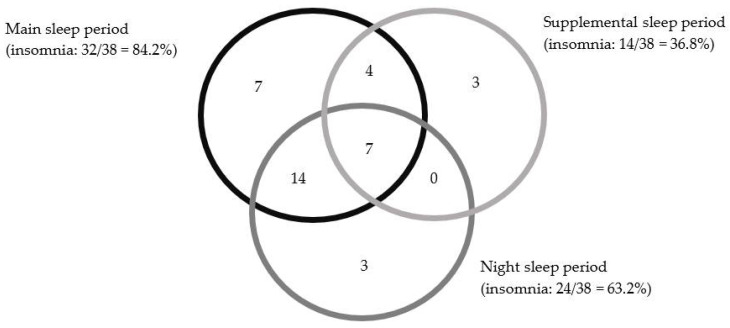
Distribution of night workers with SWD that includes insomnia symptoms (*n* = 38) across each of sleep period. *Note.* The size of circles has no meaning.

**Figure 2 brainsci-11-00928-f002:**
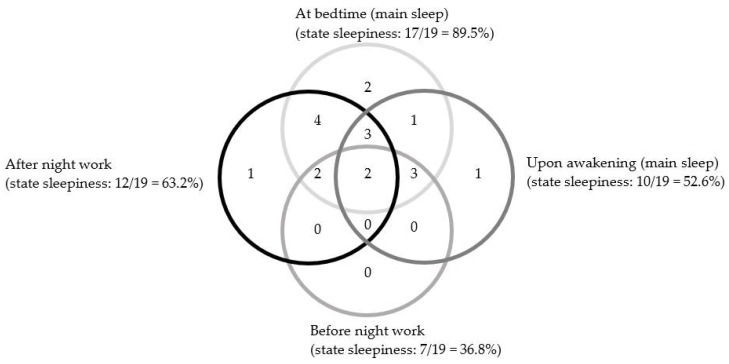
Distribution of night workers with SWD that includes sleepiness symptoms (*n* = 19) across four times. *Note.* The size of circles has no meaning.

**Table 1 brainsci-11-00928-t001:** Demographic characteristics for night workers and for day workers.

Sociodemographic	All (N = 119)	Night Workers		Day Workers		Night Workers	Day Workers
		GS (*n* = 36)	SWD (*n* = 43)	GS (*n* = 20)	Insomnia (*n* = 20)	GS vs. SWD *p*-Value ^a^	GS vs. Insomnia *p*-Value ^a^
Age in years (M, SD)	37.8 (12.1)	39.4 (14.6)	35.4 (10.4)	36.4 (12.5)	41.5 (9.1)	0.1815	0.1484
Gender (% male)	22.7%	19.4%	25.6%	30.0%	15.0%	0.5172	0.4506
Years working under this schedule (M, SD)	5.7 (6.8)	7.9 (7.2)	4.1 (4.7)	6.8 (10.6)	4.3 (4.0)	**0.0096**	0.3599
Nationality (% Canadian)	99.2%	97.2%	100.0%	100.0%	100.0%	0.4557	-
Single, separated/divorced (%)	37.8%	47.2%	30.2%	40%	35%	0.1212	0.6485
Schooling in years (M, SD)	15.2 (2.5)	14.9 (2.1)	15.5 (2.4)	15.5 (1.6)	14.4 (3.7)	0.2597	0.2035
Full time employment (%)	76.5%	83.3%	69.8%	75.0%	80.0%	0.1600	1.0000
Annual income (% participants)						0.3223	0.8330
<35,000$	19.3%	16.7%	11.6%	25%	35%		
35,000 and 50,000$	39.5%	41.7%	37.2%	45%	35%		
>50,000$	36.1%	36.1%	51.2%	20.0%	20.0%		
Refuse to answer	2.5%	5.6%	0.0%	0.0%	5.0%		
Residence (% participants)						0.0796	0.2619
Alone	18.0%	30.6%	11.6%	20.0%	5.0%		
With partner	32%	36.1%	32.6%	25.0%	25.0%		
With partner + children or family	42.0%	25.0%	48.8%	35.0%	65.0%		
Other	6.7%	8.3%	7.0%	10.0%	0.0%		

Note. M = Mean; SD = Standard deviation; GS = Good Sleep; SWD = Shift Work Disorder; ^a^ χ^2^ test *p*-value or Fisher’s exact test *p*-value when χ^2^ is not valid. *p*-values in bold indicate significant results (<0.05).

**Table 2 brainsci-11-00928-t002:** Summary of study design, tools, and procedures.

	Recruitment (15 min)	Visit 1		Observation (Two Weeks)	Visit 2 (One Hour)	Group Assignation (One Hour)
		Evaluation (Three Hours)	Observation (45 min)			
Tools used	Phone screening	SII—adapted SCID-IV and 5	Questionnaires: PSAS Consumption history DBAS-16 GCTI BDI-II STAI Work schedule satisfaction Work satisfaction DAS	Sleep Diary SSS	Feedback and sleep hygiene advice. Compensation ($50)	Algorithm based on the SWD criteria of the ICSD-III and the insomnia criteria of the DSM-5.
Objective	To determine participant eligibility	To make the diagnosis	To assess variables that might be related to SWD.	To obtain sleep variables and sleepiness measures. The sleep diary data were used to confirm if insomnia criteria were met.	Enhance compliance to study procedures	To assign participants to the good sleeper group or insomnia/SWD group.

Note. SII—adapted = Structured Insomnia Interview adapted to screen for SWD; SCID-IV = Semi-structured Clinical Interview for DSM-IV; PSAS-somatic = Predisposition Sleep Arousal Scale somatic scale; qty per week = quantity per week; PSAS = Predisposition Sleep Arousal Scale; DBAS-16 = Dysfunctional beliefs and attitudes about sleep 16-item; GCTI = Glasgow content of thoughts inventory; BDI-II = Beck Depression Inventory II; STAI = State Trait Anxiety Inventory; DAS = dyadic adjustment scale; SSS = Stanford Sleepiness Scale; SWD = Shift Work Disorder.

**Table 3 brainsci-11-00928-t003:** LsMean for sleep variables according to Work schedule (night or day workers) and sleep disorder (good sleep or SWD/insomnia).

	Night Workers		Day Workers				Night Workers	Day Workers
	GS (*n* = 36)	SWD (*n* = 43)	GS (*n* = 20)	Insomnia (*n* = 20)	ANCOVA		GS vs. SWD	GS vs. Insomnia
Sleep	LsMean (SD)	LsMean (SD)	LsMean (SD)	LsMean (SD)	F (p_global_)	F (p_interaction_)	ES (*p*-Value)	ES (*p*-Value)
**Main sleep** (in minutes)								
Main-TST	369.9 (63.1)	339.1 (61.6)	418.8 (59.1)	383.4 (62.3)	**9.13 (<0.0001)**	0.04 (0.842)	−0.56 (0.003)	
Main-TWT	43.0 (39.7)	68.7 (38.8)	41.8 (37.2)	130.1 (39.2)	**16.92 (<0.0001)**	**18.48 (<0.0001)**	0.70 (0.003)	2.38 (<0.0001)
Main-SOL	17.2 (13.1)	19.2 (12.8)	13.2 (12.3)	38.9 (12.9)	**11.20 (<0.0001)**	**24.50 (<0.0001)**	0.16 (0.485)	2.10 (<0.0001)
**Night sleep** (in minutes)								
Night-TST	443.5 (78.5)	421.1 (76.8)	420.5 (73.6)	375.8 (77.6)	**3.20 (0.01)**	0.60 (0.44)	−0.42 (0.026)	
Night-TWT	51.7 (51.0)	90.1 (49.8)	43.2 (47.8)	133.6 (50.4)	**11.00 (<0.0001)**	**7.70 (0.006)**	0.81 (0.001)	1.90 (<.0001)
Night-SOL	20.2 (18.5)	21.5 (18.1)	13.8 (17.3)	40.2 (18.3)	**5.40 (<0.0001)**	**13.70 (<0.0001)**	0.08 (0.731)	1.53 (<.0001)

Note. LsMean = Least square mean, i.e., means adjusted for age and gender; SD = standard deviation; GS = good sleep; SWD = Shift Work Disorder; ES = effect size; Main sleep = average of all main sleep periods after night work for night workers and night sleep for days workers; Night sleep = average of all night sleep periods regardless of work days or days off; TST = Total Sleep Time; TWT = Total Wake Time; SOL = Sleep Onset Latency. Day workers main sleep may include night and day sleep, if any.

**Table 4 brainsci-11-00928-t004:** Least square mean (LsMean) and standard deviation (SD) for nap and sleepiness according to work schedule (night or day workers) and sleep disorder (good sleep or SWD/insomnia).

	Night Workers		Day Workers				Night Workers	Day Workers
	GS (*n* = 36)	SWD (*n* = 43)	GS (*n* = 20)	Insomnia (*n* = 20)	ANCOVA or Linear Model		GS vs. SWD	GS vs. Insomnia
Naps and Sleepiness	LsMean (SD) or %	LsMean (SD) or %	LsMean (SD) or %	LsMean (SD) or %	F or χ^2^ (p_global_)	F or χ^2^ (p_interaction_)	ES (*p*-Value)	ES (*p*-Value)
**Naps**								
Average per day	1.27 (0.30)	1.29 (0.33)	0.98 (0.30)	0.99 (0.30)	**5.81 (<0.0001)**	0.00 (0.9781)	−1.08 (<0.0001)	
Nap-TST	108.59 (56.5)	89.31 (55.00)	91.92 (52.40)	49.31 (54.40)	**4.74 (0.0007)**	0.85 (0.3603)	−0.55 (0.0254)	
Nap-TWT	23.52 (23.80)	36.25 (23.10)	18.33 (22.00)	35.45 (22.90)	**3.56 (0.0055)**	0.17 (0.6823)	−0.14 (0.5721)	
Average per 14 days	1.45 (3.60)	2.82 (3.50)	0.36 (3.40)	0.51 (3.50)	1.92 (0.0989)	0.54 (0.4632)	−0.51 (<0.0042)	
Nap-TST on 14 days	895.54 (502.4)	720.48 (489)	154.34 (465.9)	43.34 (484.2)	**18.12 (<0.0001)**	0.08 (0.7772)	−1.55 (<0.0001)	
Nap-TWT on 14 days	160.2 (157.7)	253.51 (153.5)	31.74 (146.3)	88.52 (152)	**6.93 (<0.0001)**	0.27 (0.6074)	−1.02 (0.0001)	
**Sleepiness**								
ESS	8.2 (4.6)	8.7 (4.4)	7.8 (4.3)	9.2 (4.5)	1.36 (0.251)	0.3 (0.619)	0.21 (0.262)	
SSS	2.8 (0.72)	3.1 (0.72)	2.7 (0.72)	3.2 (0.72)	**3.96 (0.005)**	0.4 (0.539)	0.60 (0.001)	
Excessive (% yes)	14.3%	32.5%	15.0%	30.0%	11.3 (0.104)	0.0 (0.970)	0.91 (0.490)	

Note. LsMean = Least square mean, i.e., means adjusted for age and gender; SD = Standard deviation; GS = Good sleep; SWD = Shift Work Disorder; ES = effect size; Average per day = average number of additional sleep periods per day of naps; Average per 14 days = average number of additional sleep periods per 14 evaluation days; Nap-TST = average of total sleep time for all naps on day of nap in minutes; Nap-TWT = Average of Total Wake Time for all naps on day of nap in minutes; Nap-TST on 14 days = average of total wake time for all naps on 14 days in minutes; Nap-TWT on 14 days = average of total wake time for all naps on 14 days in minutes; ESS = Epworth Sleepiness Scale; SSS = Stanford Sleepiness Scale asked during the evaluation; Excessive = excessive sleepiness asked during the evaluation.

**Table 5 brainsci-11-00928-t005:** LsMean and standard deviations for psychosocial variables according to work schedule (night or day workers) and sleep disorder (good sleep or SWD/insomnia).

	Night Workers		Day Workers		Ancova (or Chi-Square ^a^)		Night Workers	Day Workers
Psychosocial Variables	GS (*n* = 36)	SWD (*n* = 43)	GS (*n* = 20)	Insomnia (*n* = 20)			GS vs. SWD	GS vs. Insomnia
	LsMean (SD)	LsMean (SD)	LsMean (SD)	LsMean (SD)	F (p_global_)	F or χ^2^ (p_interaction_)	ES (*p*-Value)	ES (*p*-Value)
**Physiological activation and lifestyle habit**								
Physiological activation (PSAS-somatic)	11.18 (3.7)	12.03 (3.6)	9.65 (3.5)	13.47 (3.6)	**3.50 (0.006)**	**4.80 (0.031)**	0.25 (0.278)	1.11 (0.001)
Smoker (% yes)	22.22%	6.98%	10%	20%	-	0.21 (0.645)	RR = 1.04 (0.785)	
Cigarettes (qty per week)	11.16 (33)	6.58 (32.2)	4.85 (31)	17.7 (32.7)	0.6 (0.702)	2.03 (0.157)	0.05 (0.795)	
Alcoholic beverages (qty per week)	2.41 (3.6)	3.18 (3.6)	3.02 (3.4)	3.79 (3.6)	1.14 (0.344)	0.00 (0.997)	0.23 (0.22)	
Caffeine beverages (qty per week)	11.89 (9)	8.36 (8.8)	7.8 (8.5)	11.67 (9)	2.21 (0.058)	4.88 (0.029)	−0.42 (0.071)	0.46 (0.151)
**Cognitive activation**								
Cognitive activation (PSAS-cognitive)	14.61 (5.3)	18.08 (5.2)	14.25 (4.9)	20.57 (5.2)	**6.10 (<0.0001)**	2.1 (0.15)	0.90 (<0.0001)	
Beliefs (DBAS-16)	3.85 (1.3)	4.68 (1.3)	3.53 (1.2)	5.91 (1.3)	**12.82 (<0.0001)**	**10.68 (0.001)**	0.70 (0.002)	1.99 (<0.0001)
Intrusive thoughts (GCTI)	40.45 (10.1)	47.56 (10.1)	37.89 (9.7)	52.14 (10.1)	**5.96 (<0.0001)**	3.11 (0.081)	0.99 (<0.0001)	
**Stimulus control behavior in bed**								
TV, phone, electronic tablets (% yes)	72.22%	76.74%	85%	75%	-	0.23 (0.635)	RR = 1.00 (0.985)	
Reading (% yes)	30.56%	48.84%	55%	65%	-	0.03 (0.873)	RR = 0.91 (0.503)	
**Emotions**								
Depression (BDI-II)	3.8 (6.5)	7.98 (6.4)	3.53 (6.1)	11.77 (6.5)	**7.85 (<0.0001)**	2.82 (0.096)	0.92 (<0.0001)	
Anxiety STATE	26.69 (9.6)	32.53 (9.4)	28.6 (9.0)	34.37 (9.6)	**3.63 (0.004)**	0 (0.985)	0.66 (0.001)	
Anxiety TRAIT	30.84 (8.2)	36.38 (8.1)	32.38 (7.7)	41.84 (8.2)	**7.07 (<0.0001)**	1.6 (0.209)	0.90 (<0.0001)	
**Sleep facilitation**								
Work satisfaction	28.25 (4.9)	26.17 (4.8)	29.20 (4.6)	26.25 (4.8)	**2.51 (0.046)**	0.23 (0.629)	−0.53 (0.005)	
Work schedule satisfaction	17.40 (4.0)	16.90 (4.0)	20.51 (3.8)	22.59 (3.9)	**8.81 (<0.0001)**	2.50 (0.117)	1.16 (<0.0001)	
Domestic and Social situation	72.59 (17.4)	69.29 (16.2)	66.48 (15.7)	68.41 (17.1)	0.39 (0.811)	0.25 (0.624)	−0.05 (0.889)	
Marital distress (DAS)	75.12 (11.8)	72.23 (12.0)	70.61 (11.5)	69.86 (12.3)	1.44 (0.22)	0.16 (0.693)	−0.19 (0.372)	
Distance from home to work (km)	13.15 (16.8)	19.20 (16.4)	11.05 (16.2)	15.42 (17.1)	1.26 (0.291)	0.33 (0.566)	0.34 (0.131)	

Note. LsMean = Least square mean, i.e., means adjusted for age and gender; SD = standard Deviation; GS = Good sleep; SWD = Shift Work Disorder; ES = Effect size; PSAS-somatic = Predisposition Sleep Arousal Scale somatic scale; qty per week = Quantity per week; PSAS-cognitive = Predisposition Sleep Arousal Scale cognitive scale; DBAS-16 = Dysfunctional beliefs and attitudes about sleep 16-item; GCTI = Glasgow content of thoughts inventory; BDI-II = Beck Depression Inventory II; STATE = State Trait Anxiety Inventory-State; TRAIT = State Trait Anxiety Inventory-Trait; DAS = dyadic adjustment scale; km = kilometers. ^a^ Chi-square tests for smoker and variables in the stimulus control behavior in bed section.

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
