# Peer review of "Psychosocial Features of Shift Work Disorder"

_brainsci, 2021, doi:10.3390/brainsci11070928_

Round 1
Reviewer 1 Report
Dear authors
This study is to show that SWD is associated with psychosocial variables based on standardized questionnaires and survey performed by experts.
- Please summarize the method with table or figure for readers to easily understand the study design. For example,
|
Tool |
Objective |
|
|
|
|
|
|
|
|
|
Authors explained the importance and necessity of integrative model for SWD in introduction. But the analyses were not done. If the integrative model were applied, this article would be better. It is necessary to show interaction or mediation.
2. Results
There is no table for general characteristics including age, sex, education, income, and so on, for participants.
It is important to show whether day and night workers had similar or different general characteristics. General characteristics may affect SWD.
There are no Table 1 and Table 3.
Reviewer 2 Report
The objective of this study is to investigate the psychosocial features of night workers with Shift Work Disorder (SWD). To conduct this study the authors compared night workers with or without SWD to the day-time workers with or without insomnia. The authors found that night worker with SWD had a lower total sleep time and higher sleepiness after main sleep periods compared to other groups. They also found that night-time workers presented with higher cognitive activation than day workers which could be a contributing factor to SWD. Night workers were found to nap more than day workers who in turn had higher sleep onset latency. Overall, these findings show the importance of evaluating all sleep periods of night workers rather than just focusing on main sleep periods.
The strengths of this study include a clear definition of each experimental group and exclusion criteria that would enhance the separation between these groups. The authors also clearly state the limitations of their study and discuss points of improvements that would make this study more representative of the general patient population. However, as the authors indicate diagnosing SWD is a real challenge. Because of its many facets, this seemingly simple disorder is complex to assess. Some of these challenges are also reflected in their finding and interpretation. We need more studies like this using advanced technology to diagnosing SWD and associated disorders. I see the following limitations which need to be addressed/discussed to publish the finding to this pilot–based study.
- This is mainly an observational pilot study with very limited clinical/mechanistic detail. Even with their observational data, the authors should not emphasize enough the significance of this study. They indicated several limitations which is important however, accentuating significance and future direction is also important.
- Also, in most of the figures/tables, they just indicate a number with statistical significance however, did not describe what it means? For example, in The Venn diagram presented in Fig.1, 7 night-time workers are common across each sleep period. However, what this means, and what is the significance in showing this in the Vann diagram. The authors should emphasize if these 7 night-time workers are some additional risks compared to the rest of the night-time workers.
- The sample size is relatively small however, the authors not extensively discussed is gender balance and representativeness. They mention that the majority of the participants were women (77.3%); however, having a more balanced sample is crucial to the representativeness of the conclusions. At least the authors should indicate if they see any gender-specific or age-specific difference.
- Most of the findings are based upon questionnaires/maintaining a diary rather than using modern devices/Apps. As authors should aware that there are many devices/apps are available for more correctly diagnostic of sleep/SWD and associated disorders and I am wondering why authors did not take advantage of these?
- The description of higher cognitive activation needs more clarity and its association with SWD.
- The manuscript has several reparative sentences/statements (word-by-word) and has some grammatical errors, which should be addressed.
Reviewer 3 Report
Dear Authors,
Generally this is an interesting work on a topic very widely discussed now in the scientific literature, as evidenced by the huge number of publications on this topic. The authors have conducted a very extensive and time-consuming study on sleep problems and daily sleepiness in shift workers. The authors used Sleep Diary to calculate the Total Sleep Time (TST), Total Wake Time (TWT), and Sleep Onset Latency (SOL) as well as main sleep, night sleep, and napping for each participant. Sleep variables were averaged out of 14 days regardless of work days or off days. In addition, they used a lot of different questionnaires for evaluation the physiological and cognitive activations, stimulus control behavior, emotions, sleep facilitation variables, work satisfaction, work schedule satisfaction, social and domestic satisfaction etc. However, I have some questions and comments to the Authors. It follows from the content, that shift workers worked in two different systems: 1) consecutive night shift: working 6 to 10 consecutive nights, followed by 8 to 4 days off respectively over 14 days (It is not clear what it means: 8-4 days off- it should be explain). 2) rotating night shift: working 6 to 10 nights, alternating with 8 to 4 days off over 14 days (the same doubts related to 8-4 days). What did the choice of this alternative depend on, whether from the employees or the employer? It is not clear whether shift workers work interchangeably in system 1 and 2 or always in the same mode. If some people work in system 1 and the other part in system 2, then in my opinion these two types of shifts should be compared, because from the work physiology point of view it is very important to assess which system is better tolerated and associated with less sleep disturbances.The habit of drinking alcohol and smoking is insufficiently described. There is no information on how much alcohol they drank and what kind (strong, wine, beer), brief remark that there were no sociodemographic diferences is not enough.
Minor remarks
Why lines numbering starts with the chapter "Measures"
line 73 unfinished phrase: where "0" ........
12 a.m - you should add midnight because it is not clear and intuitively it seems to be a noon
Table 4
- For cigarettes it should be indicated [number per day]
- Alcohol - what it means 2.41 (3.6)-it is ml or grams of alcohol, or drinks? There are no information about the kind of alcohol
- Caffeine beverages -11.89 (9) - the same doubts as related to alcohol
With regard to watching TV and reading in bed it should be specified how long it took before falling asleep
Round 2
Reviewer 1 Report
The manuscript were well updated.